# Preparation and Photocatalytic Performance of TiO_2_ Nanowire-Based Self-Supported Hybrid Membranes

**DOI:** 10.3390/molecules27092951

**Published:** 2022-05-05

**Authors:** Mohammed Ahmed Shehab, Nikita Sharma, Andrea Valsesia, Gábor Karacs, Ferenc Kristály, Tamás Koós, Anett Katalin Leskó, Lilla Nánai, Klara Hernadi, Zoltán Németh

**Affiliations:** 1Faculty of Materials Science and Engineering, University of Miskolc, H-3515 Miskolc, Hungary; kemahmed@uni-miskolc.hu; 2Polymers and Petrochemicals Engineering Department, Basrah University for Oil and Gas, Basrah 61004, Iraq; 3Advanced Materials and Intelligent Technologies Higher Education and Industrial Cooperation Centre, University of Miskolc, H-3515 Miskolc, Hungary; nikita_sh18@yahoo.in; 4European Commission, Joint Research Centre (JRC), 21027 Ispra, Italy; andrea.valsesia@ec.europa.eu; 5MTA-ME Materials Science Research Group, ELKH, H-3515 Miskolc, Hungary; femkg@uni-miskolc.hu; 6Institute of Mineralogy and Geology, University of Miskolc, H-3515 Miskolc, Hungary; askkf@uni-miskolc.hu; 7Institute of Energy and Quality Affairs, University of Miskolc, H-3515 Miskolc, Hungary; koos.tamas@uni-miskolc.hu (T.K.); kkklesko@uni-miskolc.hu (A.K.L.); 8Institute of Physical Metallurgy, Metal Forming and Nanotechnology, University of Miskolc, H-3515 Miskolc, Hungary; nanai.lilla@student.uni-miskolc.hu

**Keywords:** self-supported membranes, scanning electron microscopy, photocatalysis, organic dye decomposition

## Abstract

Nowadays, the use of hybrid structures and multi-component materials is gaining ground in the fields of environmental protection, water treatment and removal of organic pollutants. This study describes promising, cheap and photoactive self-supported hybrid membranes as a possible solution for wastewater treatment applications. In the course of this research work, the photocatalytic performance of titania nanowire (TiO_2_ NW)-based hybrid membranes in the adsorption and degradation of methylene blue (MB) under UV irradiation was investigated. Characterization techniques such as scanning electron microscopy (SEM), transmission electron microscopy (TEM), energy-dispersive X-ray spectroscopy (EDS), X-ray powder diffractometry (XRD) were used to study the morphology and surface of the as-prepared hybrid membranes. We tested the photocatalytic efficiency of the as-prepared membranes in decomposing methylene blue (MB) under UV light irradiation. The hybrid membranes achieved the removal of MB with a degradation efficiency of 90% in 60 min. The high efficiency can be attributed to the presence of binary components in the membrane that enhanced both the adsorption capability and the photocatalytic ability of the membranes. The results obtained suggest that multicomponent hybrid membranes could be promising candidates for future photocatalysis-based water treatment technologies that also take into account the principles of circular economy.

## 1. Introduction

Over the past decade, air and water pollution has been on an upward trend, causing concern in everyday life around the world. In general, the industrial sectors accumulate large quantities of wastewater during manufacturing processes, which contains various chemical components such as pharmaceutical residues, dyes or heavy metals and salts [1] that are costly to neutralize and remove. Semiconductors and their composites have been studied in depth to combat the problems associated with waste generation. They include TiO_2_ [2], ZnO [3], WO_3_ [4], Bi-based [5] and Ag-based [6] materials, etc. [7].

Titanium dioxide (TiO_2_) is one of the most widely used semiconductor for photocatalytic applications due to its advantageous properties (optical and electronic properties, chemical stability, low cost, lack of toxicity) [8]. However, its practical application is limited, as it can only be effectively excited under UV light due to its wide band gap. There have been numerous attempts to enhance the photocatalytic activity and excitability of titanium dioxide, for example, by synthesizing TiO_2_ with various morphologies [9], modifying it with noble metals [10,11], doping it with various elements [12,13] and preparing composites with graphene [14], graphene oxides [15] and other semiconductors [16].

Copper and copper oxide are widely used as dopants on titanium dioxide materials’ surfaces to enhance their photocatalytic performance by narrowing the bandgap and improving electron–hole separation with photoexcitation, as reported in [17,18]. Variations of iron oxides are used to modify the properties of titanium dioxide, due to their small band gap [19]. With the combination of iron oxide and titanium dioxide, shallow trap sites appear between the conduction band and the valance band that leads to a reduction in the band gap energy of TiO_2_ [20]. Furthermore, the radius of Fe^2+^ and Fe^3+^ ions is smaller than that of Ti^4+^; thus, iron ions can diffuse into the TiO_2_ lattice to substitute TiO_2_ [21]. In addition, iron oxides (magnetite, i.e., Fe_3_O_4_ and maghemite, i.e., Fe_2_O_3_) have special magnetic properties in the nanoscale range [22]. These ferromagnetic properties can be beneficial to improve the recyclability of photocatalysts and prevent particles from clumping together [23].

Many authors have reported that enhanced photocatalytic properties are expected from morphological modifications of TiO_2_-based photocatalytic nanomaterials, such as one-dimensional TiO_2_ nanostructures, including TiO_2_ nanowires (TiO_2_ NW) [19,24], TiO_2_ nanotubes (TiO_2_ NT) [25,26], TiO_2_ hollow spheres (TiO_2_ HS) [27] and TiO_2_ hollow fibers (TiO_2_ HF) [28]. These morphologies have a high specific surface area as well as other beneficial properties, as they can decrease the hole–electron recombination rate and increase the interfacial charge transfer rate [29,30]. TiO_2_ NW have excellent mechanical stability and chemical and physical properties. For the synthesis of TiO_2_ NW, mostly hydrothermal [31] and solvothermal methods [32] are used. Previous scientific results indicate that TiO_2_ nanowires have outstanding photocatalytic efficiency [31,32,33].

Photocatalytic membranes (PM) offer a great potential alternative for economical and eco-friendly treatments of wastewater, based on the combination of membrane filtration and photocatalysis [34,35]. Membrane filtration is widely used for drinking water treatment and wastewater treatment [36,37] due to its simple operation and effective removal of various types of pollutants. Using this technology, most organic and small amounts of inorganic substances can be decomposed using solar energy to reduce the harm of pollutants. There are various types of materials that can be used as membranes, such as natural polymers (cellulose [38,39]), green porous nano-membranes [40], graphene [41], carbon nanotube [42], ceramic [43] or synthetic polymer-based (polyvinyl alcohol (PVA), polyacrylonitrile (PAN), polyvinylidene fluoride (PVDF) etc.) [44] membranes [45]. TiO_2_ nanowire membranes have been investigated as mechanical microfilters and as photocatalysts to degrade pharmaceutical residues such as trimethoprim and other organic materials including dyes, phenol and humic acid under UV irradiation [46]. A. Hu et al. investigated various types of TiO_2_ nanostructured membranes, though the properties of CuO- and Fe_2_O_3_-modified TiO_2_ photocatalytic membranes were not explored.

Cellulose-based materials are promising substances that can partly or totally replace synthetic fibers as filters in masks [47] and can be used as membranes or support for other materials and to remove oil and heavy metal ions during water treatment [48]. Furthermore, cellulose has unique features in comparison to the usual supports and thus allows nanoparticles stability, reactivity, recyclability and prevents nanoparticle aggregation. Cellulose also represents a sustainable alternative to known methods with the aforementioned properties. CdS and TiO_2_ nanoparticle–nanocellulose hybrid composites have been used as photocatalysts for a model pollutant, i.e., methyl orange degradation. In the CdS case, 82% degradation efficiency was achieved after 90 min irradiation, and the material was reusable up to five times. [49]. A TiO_2_–cellulose hybrid composite was 20% more efficient in degrading methylene orange than pure TiO_2_ after 20 min of UV irradiation [50]. In a recent study, it was found that the photocatalytic degradation rate of TiO_2_–cellulose was much higher (99.72%) than that of bare TiO_2_ (69.18%) under UV irradiation for 30 min, because cellulose served as a support for TiO_2_ nanoparticles distribution and also promoted the adsorption of methyl orange molecules [51]. In this work, the aim was to develop a “green chemistry” solution to industrial wastewater effluents. Cellulose-based membranes were chosen over other materials due to their advantageous properties such as lack of toxicity, low cost, biodegradability and eco-friendliness. Other than this, cellulose membranes have excellent specific surface area, adjustable surface chemistry, hydrophilicity and mechanical strength [52]. 

In the current paper, we successfully developed titanium dioxide nanowire (TiO_2_ NW)-based hybrid membranes. The surface of TiO_2_ NW was decorated with iron oxide (Fe_2_O_3_) and copper oxide (CuO) nanoparticles to improve the photocatalytic performance of the as-prepared composites. Furthermore, cellulose fibers were applied as a reinforcement filler material to prepare self-supported hybrid membranes. The photocatalytic properties of the membranes were investigated against methylene blue decomposition under UV irradiation. The results showed that nanofiber-based hybrid membranes can provide an excellent alternative as environmentally friendly solutions for wastewater treatment requiring the degradation of organic pollutants.

## 2. Results

### 2.1. HRTEM and EDS Analysis of TiO_2_ NW@Fe_2_O_3_ and TiO_2_ NW@CuO Nanocomposites

Heat-treated nanocomposite samples were investigated by the HRTEM technique. Figure 1 shows HRTEM micrographs of the prepared TiO_2_ NW@Fe_2_O_3_ and TiO_2_ NW@CuO nanocomposites. These images revealed that the fabrication of both nanocomposites was successful, although different nanocomposite structures were observed during TEM. The HRTEM images showed that inorganic nanoparticles (Fe_2_O_3_ and CuO) were attached to the surface of TiO_2_ NW. Figure 1a,b show that Fe_2_O_3_ and CuO nanoparticles adhered on TiO_2_ NW, respectively, and segregated particles could not be observed. 

Furthermore, from the analysis of the HRTEM images, the average particle sizes of Fe_2_O_3_ and CuO nanoparticles were calculated using the iTEM software (Olympus Soft Imaging Solutions). The average particle size of these components was determined by measuring 100 individual particles in both samples. Based on these calculations, it was found that Fe_2_O_3_ nanoparticles had a diameter in the range of 20–30 nm, as can be seen in Figure 1a, while the average diameter of CuO nanoparticles was 2–3 nm, as shown in Figure 1b.

EDS analysis was performed to determine the elements in the as-prepared nanocomposites. Figure 2a,b show the EDS spectra and confirmed that the most significant signals originated from carbon (C), oxygen (O), titanium (Ti), potassium (K), copper (Cu) and iron (Fe). The presence of TiO_2_ NW, Fe_2_O_3_ and CuO in the samples was confirmed by the Ti, Fe, Cu and O peaks, as shown in the spectra (Figure 2), while in the case of the TiO_2_ NW@Fe_2_O_3_ nanocomposite sample, the Cu peak originated from the sample holder (a lacey Cu grid), and the peak of K was due to the residual KOH that was used during the preparation of the TiO_2_ NW. 

### 2.2. SEM and EDS Analysis of the Hybrid Membranes 

In order to gain information about the surface morphology of the as-prepared hybrid membranes, SEM analysis was performed. The surface nature and morphology of a neat cellulose membrane (Figure 3a), the TiO_2_ NW (Figure 3b) and the as-prepared TiO_2_ NWs@Fe_2_O_3_/cellulose (Figure 3c,d) and TiO_2_ NWs@CuO/cellulose hybrid membranes (Figure 3e,f) were characterized by SEM. The SEM images in Figure 3a show cellulose wires with an average diameter of 1–5 µm, while the TiO_2_ NW had an average diameter of 10–15 nm, as observed in Figure 3b. Figure 3c,e show the produced self-supported hybrid membranes, and the SEM images in Figure 3d,f demonstrate the microstructure of TiO_2_ NWs@Fe_2_O_3_/cellulose (d) and TiO_2_ NWs@CuO/cellulose (f) hybrid membranes. In the TiO_2_ NWs@Fe_2_O_3_/cellulose membrane, the Fe_2_O_3_ particles stuck together resulting in bigger agglomerates and increased pore size.

To determine the elemental composition and confirm the presence of Fe_2_O_3_ and CuO nanoparticles in the as-prepared hybrid membranes, EDS analysis was performed for each sample. The data of the EDS analysis in Table 1 revealed the atomic percentages (at%) of the detected elements in the samples. The most significant signals originated from carbon (C), oxygen (O) and titanium (Ti), confirming the presence of cellulose and TiO_2_ NW in the hybrid membranes. Furthermore, iron (Fe) and copper (Cu) signals were detected, which related to the Fe_2_O_3_ and CuO nanoparticles. Other elements were also observed, such as sodium (Na), potassium (K) and fluorine (F), which originatied from the preparation procedure of TiO_2_ NW. The results of EDS analysis from HRTEM and SEM showed good correlations.

### 2.3. XRD and Specific Surface Area Analysis of the Hybrid Membranes

In order to determine the degree of crystallization of the as-prepared nanocomposites and membranes and to identify and describe the crystal structure of these materials, XRD analysis was performed. As can be seen in Figure 4a, the diffraction peaks at the angles 2(θ°) of 24.1°, 33.2°, 35.6°, 40.8°, 49.5°, 54.1°, 57.3° and 62.6° correspond to the reflections from (012), (104), (110), (113), (024), (116), (212), respectively. All of these peaks originated from hematite (α-Fe2O3) (JCPDS 33-0664). The diffraction peaks at the angles 2(θ°) of 25.2°, 36.2°, 37.5°, 38.5°, 48.0°, 54.3° and 56,6° correspond to the (101), (103), (004), (112), (200), (105), (211) and (118) reflections, respectively, and relate to anatase phase of the TiO_2_ NW (JCPDS 21-1276). Furthermore, the diffraction peaks at 11.9°, 24.2°, 43.1°, 60.1°, correspond to the crystal phase (200), (002), (602), (610) of potassium titanium oxide K_2_Ti_6_O_13_ (PDF no. 40-0403). All these results are in agreement with previous reports [53]. The diffraction peaks of cellulose (PDF no. 03-0289) could be observed at angles 2(θ°) of 16.3°, 22.7° and 34.5°, corresponding to the crystal planes of cellulose at (110), (110), (200) and (400) reflections, respectively.

Figure 4b shows the XRD analysis of the TiO_2_ NWs@CuO/cellulose hybrid membrane. As shown, the diffraction peaks located at the angles 2(θ°) of 33.1°, 35.7°, 38.1°, 47.1°, 49.4°, 53.2°, 58.6°, 62.2°, 67.3 ° and 68.1° correspond to monoclinic crystal phases of the reflections (110), (002), (200), (112), (202), (020), (202), (113), (022), (220), related to copper oxide (CuO) (JCPDS card number 45-0937), which showed correlations with earlier published results [54].

The pure materials, the membranes and the hybrid membranes were also characterized by the N_2_ adsorption technique to determine their specific surface areas and pore diameters, as can be seen in Table 2. It was found that there was no significant difference between the surface areas of the as-prepared hybrid membranes. Both hybrid membranes had a specific surface area of approx. 120 m^2^/g.

### 2.4. Photocatalytic Efficiency of the Hybrid Membranes

The photodegradation efficiency of the synthesized membranes was tested using methylene blue dye as a model pollutant, under UV light irradiation. The results revealed the almost complete removal of MB. Approx. 90% of the initial MB dye was decomposed by the hybrid membrane containing a photocatalyst, i.e., the TiO_2_ NW@Fe_2_O_3_ and TiO_2_ NW@CuO nanocomposites. Figure 5 presents the removal efficiency of MB of the TiO_2_ NW@Fe_2_O_3_/cellulose and TiO_2_ NW@CuO/cellulose hybrid membranes under UV light.

It is important to note here that only the hybrid membranes are shown because it was reported that structure deformations appear in pure cellulose exposed to UV light. This is the reason why we applied a lower amount of cellulose in the hybrid membranes. The high removal of MB by the membranes reflects their potential to degrade organic dye molecules. The excellent degradation efficiency could be attributed to the synergistic effects of the TiO_2_ NW with nanoparticles (Fe_2_O_3_, CuO) on their surface and the adsorption of MB on cellulose. As the present study was carried out under UV light, significant differences could be observed under visible light; therefore, we will explore this possibility in a further study. In addition, since pure cellulose does not contain any photocatalyst, a pure cellulose membrane is not expected to possess photocatalytic activity. However, its adsorption capacity can be non-negligible; therefore, we performed adsorption studies for 2 h of this and other membranes, as shown in Figure 6. Since a small amount of cellulose was used in the hybrid membranes for the reason mentioned above and the adsorption capacities appeared to be similar for all materials, it can be concluded that titanate alone is capable of adsorbing MB molecules. In the literature, several works have reported enhanced adsorption of different organic pollutants onto Fe_2_O_3_ and CuO, making these compounds promising components of membranes for wastewater treatment [19,55].

## 3. Materials and Methods

### 3.1. Materials

Titanium dioxide (TiO_2_) (P25), anatase nanopowder (with an average diameter of 21 nm), ammonium hydroxide (NH_4_OH, 25%) and sodium hydroxide (NaOH) were purchased from Sigma Aldrich (Budapest, Hungary). Hydrochloric acid (HCl, 37 %) and copper (II) acetate monohydrate (Cu(OOCCH_3_)_2_ × H_2_O) were obtained from VWR Chemicals (Debrecen, Hungary). Potassium hydroxide (KOH) was purchased from Thomasker Co., (Budapest, Hungary). Iron chloride hexahydrate (FeCl_3_ × 6H_2_O) was purchased from Scharlab, (Debrecen, Hungary). Cellulose originated from DIPA Ltd. (Miskolc, Hungary). A polyvinylidene (PVDF) filter membrane with pore size of 0.1 µm and diameter of 47 mm (Durapore-VVLP04700) was used for hybrid membranes’ preparation. 

### 3.2. Synthesis of TiO_2_ Nanowires (TiO_2_ NW)

Recently, we showed the preparation of TiO_2_ NW using the so-called solvothermal process [56]. In brief, a homogeneous TiO_2_ suspension was transferred into a Teflon^®^-lined autoclave. The autoclave was kept in a dryer at 160 °C for 24 h. The as-prepared TiO_2_ NWs were washed with 0.1 M HCl and deionized water until a neutral pH was reached. The products were dried and calcined at 500 °C for 1 h. 

### 3.3. Synthesis of TiO_2_ NW@Fe_2_O_3_/Cellulose Membranes

For the synthesis of the hybrid membrane, firstly, a TiO_2_ NW@Fe_2_O_3_ nanocomposite was prepared. The calculated amount of FeCl_3_ × 6H_2_O precursor was dissolved in 100 mL of distilled water to obtain a homogeneous solution. Then, 0.95 g of previously prepared TiO_2_ NWs was added to the solution and stirred for 1 h, then transferred to the autoclave for 9 h at 90 °C. The product obtained was washed with 0.1 M NaOH to adjust the pH to 7, dried for 12 h at 50 °C and then calcinated for 2 h at 500 °C using a static furnace. The load of the Fe_2_O_3_ nanoparticles in the final composition was 5 *w/w* %. In the next step, 0.2 g of the as-prepared TiO_2_ NW@Fe_2_O_3_ nanocomposite powder was dispersed in 100 mL of distilled water for 1 h, then 5 g of cellulose solution (1 *w/w* %) was added to the solution and stirred for 1 h. Finally, the preparation of the cellulose-based hybrid membranes was accomplished by vacuum filtration through a PVDF membrane (total mass of 250 mg/membrane), followed by drying in a furnace for 30 min at 40 °C. 

### 3.4. Synthesis of TiO_2_ NW@CuO/Cellulose Membranes

In the same way, a calculated amount of (Cu(CH_3_COO)_2_ × H_2_O) was dissolved in 100 mL of EtOH and left under vigorous stirring for 30 min to ensure complete dissolution. Then, 0.95 g of TiO_2_ NW was added directly to the solution, which was kept under vigorous stirring for 1 h. The mixture was poured into an autoclave and placed in a static furnace at 150 °C for 12 h. The final product was collected and washed using vacuum filtration and calcinated for 2 h at 500 °C. The load of the CuO nanoparticles in the final composition was 5 *w/w* %. To prepare the TiO_2_ NW@CuO/cellulose membranes, 0.2 g of the above prepared composite was dipped into 100 mL of EtOH for 1 h, then 5 g of cellulose (1%) was added to the solution for another 1 h, and finally, the membrane was obtained by vacuum filtration using a PVDF membrane (total mass of 250 mg/membrane), followed by drying in a furnace for 30 min at 40 °C.

### 3.5. Characterization Techniques

For the qualitative characterization, high-resolution transmission electron microscopy (FEI Tecnai G^2^ F20 HRTEM, Hillsboro, OR, USA) was used to analyze the morphology of the synthetized TiO_2_ NW@Fe_2_O_3_ and TiO_2_ NW@CuO nanocomposites. To prepare the samples for HRTEM, the nanocomposites were dispersed in ethanol and sonicated for 5 min. On a Cu TEM-grid (300-mesh copper grids, lacey carbon, Ted Pella Inc., Redding, CA, USA), we placed a droplet of each suspension. The diameter of the materials was determined using the ImageJ software, utilizing the HRTEM images and the original scale bar. To determine the elemental composition of the TiO_2_ NW@Fe_2_O_3_ and the TiO_2_ NW@CuO nanocomposites, energy-dispersive X-ray spectroscopy (EDS; AMETEK Inc., Berwyn, PA, USA; active area 30 mm^2^) coupled to HRTEM was applied. 

The crystal structure of the TiO_2_ NW@Fe_2_O_3_/cellulose and TiO_2_ NW@CuO/cellulose membranes was determined using X-ray powder diffraction (XRD) (Bruker D8 Advance diffractometer, Billerica, MA, USA) at (CuK α = 0.15418 nm; 40 kV and 40 mA) in parallel beam geometry (Göbel mirror) with a position-sensitive detector (Vantec1, Springfield, IA, USA; 1°opening). On top-loaded specimens in zero-background Si sample containers, measurements were made in the 2–80° (2 Theta) range using a 0.007° (2 Theta)/14 s goniometer speed. 

The surface morphology of the as-prepared TiO_2_ NW@Fe_2_O_3_/cellulose and TiO_2_ NW@CuO/cellulose membranes was investigated by scanning electron microscopy. SEM and EDS spectroscopy was carried out in a Nova 600i Nanolab (Thermofisher, Eindhoven, The Netherlands) equipped with an EDS system for elemental analysis (EDAX Inc., Mahwah, NJ, USA). The EDS system mounted an Octane Elect Plus X-rays detector. Typical EDS maps and spectra were acquired using acceleration voltage values between 10 kV and 25 kV, with take-off angle of 35° and Dwell Time of 200 ms. For SEM analysis, the powders were deposited on a sticky carbon tape. Both powders and membranes were imaged directly without any conductive coating. The typical SEM working distance was 5 mm, and the acceleration voltage ranged from 2 kV up to 25 kV, depending on image quality and charging conditions.

To determine the surface area of the raw materials, the nanocomposites and the hybrid membranes, nitrogen adsorption–desorption experiments were carried out at 77 K to determine the Brunauer–Emmett–Teller [24] (BET) specific surface area using an ASAP 2020 instrument (Micromeritics Instrument Corp., Norcross, GA, USA).

### 3.6. Photocatalytic Experiments

The photocatalytic activity of the membranes was evaluated by studying the degradation of methylene blue as a model pollutant in an aqueous solution under UV light irradiation. The membranes were dipped into 100 mL of 0.03 mM MB solution and kept in the dark for two hours to attain the adsorption–desorption equilibrium. After this, the solution containing the pollutant and the photocatalyst (hybrid membranes) was exposed to UV-A lamps at a power between 300 and 500 W (Cosmedico N 400 R7S) for 60 min. The samples were withdrawn at regular time intervals and analyzed using a UV–Vis spectrophotometer (BEL UV-M51). The removal efficiency of the membranes was measured by recording the absorbance at 664 nm, and the degradation efficiency (% *deg.*) was calculated using the following formula:$$\%\;deg.=\frac{\left(c_{0}-c_{t}\right)}{c_{0}}\times 100$$
where, *c*_0_ is the initial concentration at time *t* = 0, *c_t_* is the concentration at time “*t*”, and % *deg.* is the photodegradation efficiency of the materials in relation to MB removal.

## 4. Conclusions

Herein, we presented the synthesis of two types of novel TiO_2_ nanowire-based hybrid membranes and studied their adsorption properties and photocatalytic performance in the decomposition of methylene blue under UV light. The as-prepared TiO_2_ NW@Fe_2_O_3_/cellulose and TiO_2_ NW@CuO/cellulose membranes were characterized by TEM, SEM, EDS and XRD. The results clearly demonstrated that not only the preparation of raw TiO_2_ NW, Fe_2_O_3_ and CuO decorated TiO_2_ NW, but also the production of hybrid membranes was successful. 

Comparing the results of photocatalysis, it was found that both types of hybrid membranes showed outstanding performance in removing MB in only 60 min of UV irradiation. The photocatalytic degradation efficiency in MB removal of TiO_2_@Fe_2_O_3_/cellulose was 88%, while that of TiO_2_@CuO/cellulose membrane was up to 90%.

Since we applied UV light in this study, a significant degradation could presumably be observed under visible light irradiation; therefore, we intend to explore this possibility in a future study. Furthermore, we are planning to submit a further study in the near future regarding the microbiological and toxicological properties of the hybrid membranes here presented. It is believed that by exploiting the advantageous properties of the hybrid membrane-based water treatment technologies and solutions presented here, new and sustainable strategies could be implemented for photocatalyst-based water treatment technologies.

## Figures and Tables

**Figure 1 molecules-27-02951-f001:**
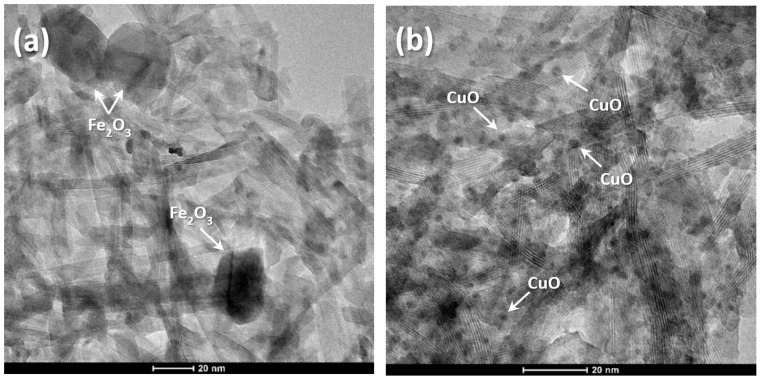
HRTEM images of TiO_2_ NW@Fe_2_O_3_ (**a**) and TiO_2_ NW@CuO (**b**) nanocomposite samples.

**Figure 2 molecules-27-02951-f002:**
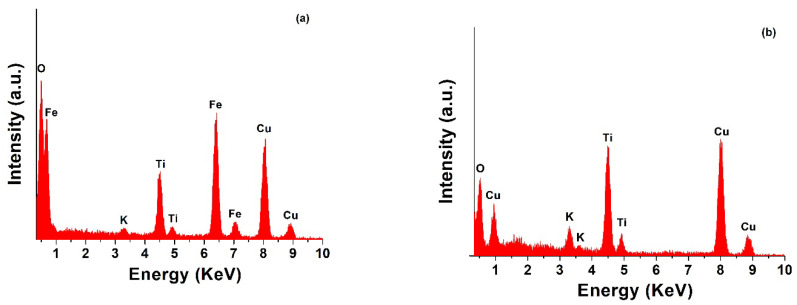
EDS spectra of TiO_2_ NW@Fe_2_O_3_ (**a**) and TiO_2_ NW@CuO (**b**) nanocomposites.

**Figure 3 molecules-27-02951-f003:**
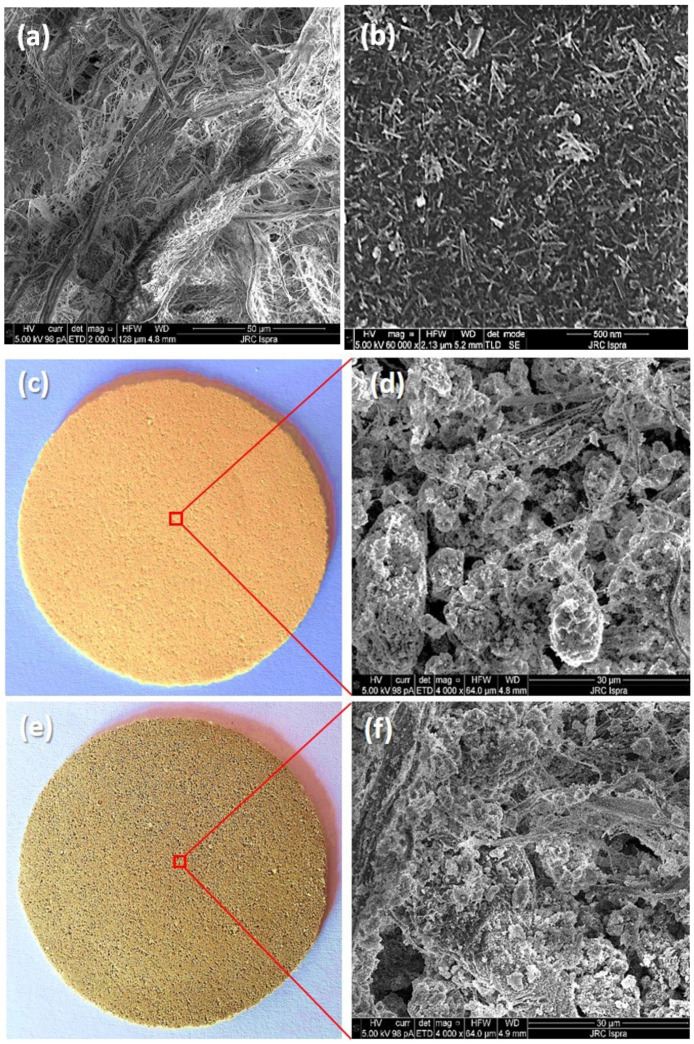
SEM images of neat cellulose (**a**), TiO_2_ NW (**b**); photograph and SEM images of TiO_2_ NW@Fe_2_O_3_ (**c**,**d**) and TiO_2_ NW@CuO (**e**,**f**) nanocomposite samples.

**Figure 4 molecules-27-02951-f004:**
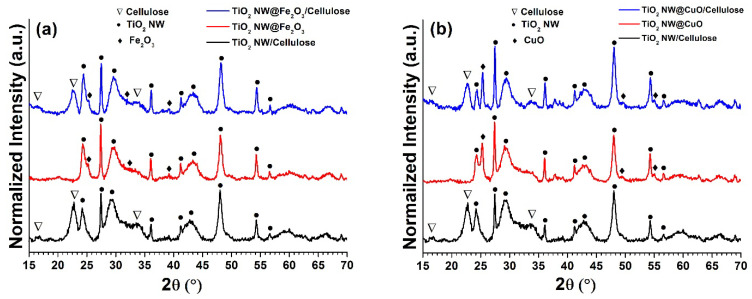
XRD of TiO_2_ NW@Fe_2_O_3_/cellulose (**a**) and TiO_2_ NW@CuO/cellulose (**b**) hybrid membranes.

**Figure 5 molecules-27-02951-f005:**
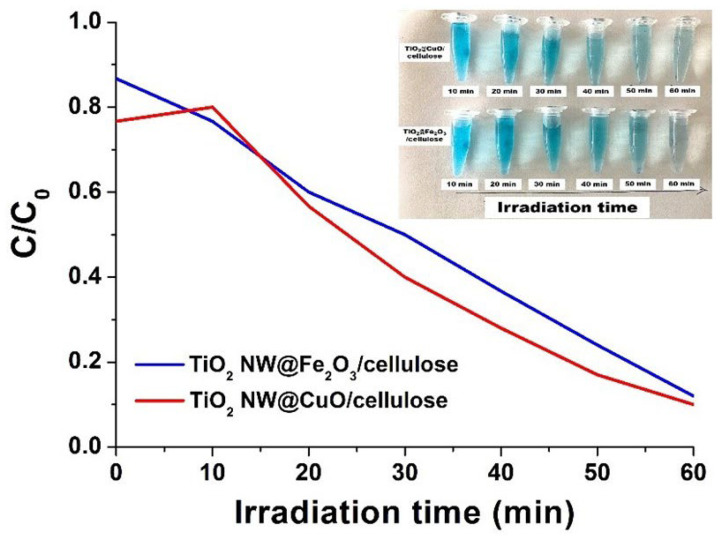
Photocatalytic performance of TiO_2_ NW@Fe_2_O_3_/cellulose (blue curve) and TiO_2_ NW@CuO/cellulose (red curve) hybrid membranes in MB degradation.

**Figure 6 molecules-27-02951-f006:**
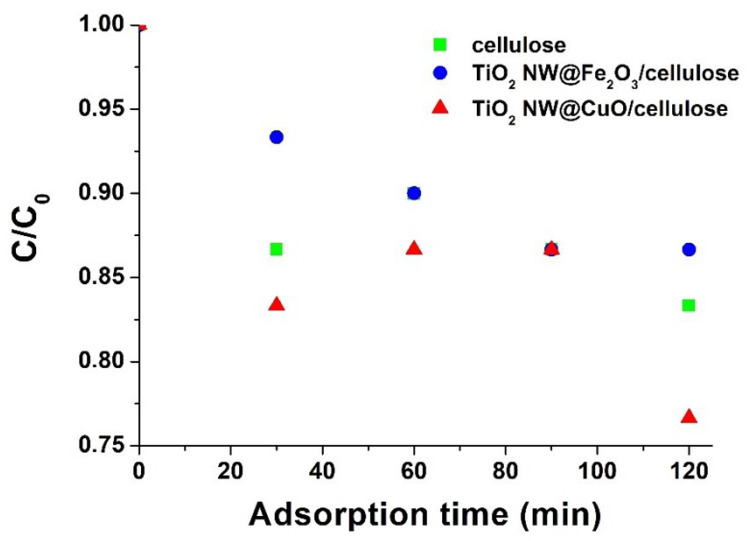
Adsorption capacity of pure cellulose (green marks) as well as TiO_2_ NW@Fe_2_O_3_/cellulose (blue marks) and TiO_2_ NW@CuO/cellulose (red marks) hybrid membranes in MB adsorption tests.

**Table 1 molecules-27-02951-t001:** Elemental (EDS) analysis of the raw materials (cellulose and TiO_2_ NW) and hybrid membranes.

Sample Name	C	O	Ti	Na	F	K	Fe	Cu
Cellulose	45	55	-	-	-	-	-	-
TiO_2_ NW	-	64	36	-	-	-	-	-
TiO_2_ NW@Fe_2_O_3_/cellulose	20	50	23	3	-	3	1	-
TiO_2_ NW@CuO/cellulose	25	58	11	-	3	2	-	1

**Table 2 molecules-27-02951-t002:** Specific surface areas of the pure materials and the hybrid membranes.

Sample Name	Surface Area (m^2^/g)	Pore Diameter (nm)
TiO_2_ NW	168	-
cellulose membrane	6	18
Fe_2_O_3_	62	-
CuO	141	-
TiO_2_ NW@Fe_2_O_3_	146	-
TiO_2_ NW@CuO	139	-
TiO_2_ NW@Fe_2_O_3_/cellulose	122	16
TiO_2_ NW@CuO/cellulose	117	16

## Data Availability

Not applicable.

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
