# Peer review of "Preparation and Photocatalytic Performance of TiO2 Nanowire-Based Self-Supported Hybrid Membranes"

_molecules, 2022, doi:10.3390/molecules27092951_

Round 1

Reviewer 1 Report

The manuscript “Preparation and photocatalytic performance of TiO2 nanowire-based self-supported hybrid membranes”  by Shehab et al. reports the synthesis of two novel TiO2 nanowire-based hybrid membranes (TiO2 NWs@Fe2O3/cellulose and TiO2 NWs@CuO/cellulose hybrid membranes) and their characterization. Their photocatalytic performances in the degradation of methylene blue (MB) under UV irradiation are also reported, towards new hybrid membranes for wastewater treatment and water remediation. The hybrid membranes showed removal of MB dye with degradation efficiency of 90% in 60 minutes. The authors attributed a higher efficiency to the presence of binary components of the membrane that enhanced both the adsorption capability and photocatalytic ability of the membranes.

Except for the points mentioned hereafter (in need of corrections), the article is suitably written and the work is interesting. However, the manuscript is not yet ready for publication: hereafter are listed important points that need particular attention in the corrections.

1) The work is based on the novel combination of cellulose and TiO2 / Fe2O3 and CuO as supported hybrid catalysts. However, the explaination of the choice made for a BIODEGRADABLE support (cellulose) to support NON-biodegradable TiO2 nanowires for water treatment and remediation is missing in the introduction. There is this statement (line 98) “Cellulose fibers was applied as a reinforcement filler material to prepare self-supported hybrid membranes. Photocatalytic properties of the membranes were investigated against methylene blue decomposition under UV irradiation. The results showed that nanofiber-based hybrid membrane technology can provide an excellent alternative for environmentally friendly solutions for waste-water treatment and such as degradation of organic pollutants.” This statement is not convincing and does not answer the above-mentioned question. Please add in introduction a full explanation, supported by literature examples and data (convincing numbers).

2) Studies to verify that the new hybrid membranes are not leaching photocatlysts in the treated water are missing. Please add and conclude.

3) A summary table of previous report on similar photocatalysts (keeping in mind that Fe2O3 nanoparticles have a diameter in the range of 20-30 nm, while the average diameter of CuO nanoparticles was 2-3 nm) is missing in the discussion. This is necessary in order to benchmark the newly developed systems and demonstrate the superior use of cellulose as a support.

4) Typo to correct: Line  308 « as shown in Fig. 6.” Should be figure 5.

Reviewer 2 Report

The TiO2 NW@Fe2O3/cellulose and TiO2 NW@CuO/cellulose were synthesized and used as effective photocatalysts for MB removing from water. The subject is interesting and useful characterizations along with an acceptable organization can be found in the manuscript. Hence, the manuscript can be suggested for publication. However, there are some points which should be addressed and/or discussed in the revised version, as mentioned below:

  1. In Figure 4, the phase formation and crystalline orientation of all peaks should be given in the figure.

  1. The porous material can show high adsorption property. The adsorption results in dye removing from the environment. But, it is completely different from the photocatalytic degradation. So, the contribution of the adsorption and degradation should be determined separately. Performing the experiments in dark can help in this regard.

  1. It has been mentioned that “There have been numerous attempts to enhance the photocatalytic activity and excitability of titanium dioxide, for example, synthesizing TiO2 with various morphologies [9], modifying with noble metals [10][11], doping with various elements [12][13] and preparing composites with other semiconductors[14].”. But, one of the recent methods for enhancing the photocatalytic activity of TiO2 is its heterogeneous formation with graphene. See, for example, [Photocatalytic reduction of graphene oxide nanosheets on TiO2 thin film for photoinactivation of bacteria in solar light irradiation] and [Adverse effects of graphene incorporated in TiO2 photocatalyst in minuscule animals under solar light irradiation]. This should be mentioned in the introduction section for further completion.

  1. TiO2 nanowire membranes were previously used for water purification. See, for example, [Journal of Hazardous Materials Volume 189, Issues 1–2, 15 May 2011, Pages 278-285] and [Journal of Membrane Science Volume 472, 15 December 2014, Pages 167-184]. The advantages of this work as compared to the previous ones should be further highlighted in the introduction section.

  1. Along with Table 2, the adsorption-desorption curves should be also given in the manuscript. In addition, the shape of the curves can determine the general shape of the porous. This should be discussed in the revised version using suitable supports.     

  1. It has been stated that “With the combination of iron oxide and titanium dioxide, shallow trap sites appearing between the conduction band and valance band that lead to reduce the band gap energy of TiO2 [18].”. It is right. However, as another effective mechanism, the smaller radius of Fe3+ than that of Ti4+ can promote diffusion of iron ions into the TiO2 lattice for substitution in the TiO2 (see, for example, [Thickness dependent activity of nanostructured TiO2/α-Fe2O3 photocatalyst thin films]). This should be mentioned in the revised version.

  1. It has been mentioned that “With the combination of iron oxide and titanium dioxide, shallow trap sites appearing between the conduction band and valance band that lead to reduce the band gap energy of TiO2”. This statement can be further supported by the following article: [Applied Catalysis A: General 369 (2009) 77-82].

  1. Could the authors comment on the stability in the photocatalytic performance of the samples after many cycles?

  1. Could the authors comment on the effect of pH on the photocatalytic performance of the samples?

  1. Some important membranes have been listed in the introduction section. This can be further completed by mentioning green porous membranes (see, for example, [Journal of Hazardous Materials 423 (2022) 127130]).  

Round 2

Reviewer 1 Report

Dear Authors

I would like to thank you for your careful and detailed replies and corrections.

I will recommand publication of the corrected version. However, I have some remaining comments for you.

a) I will drop my point Q3 for this occasion.

b) My point Q1. After reading your reply, I maintain that I am not convinced, for future application, by the choice of a BIODEGRADABLE support (cellulose) to support NON-biodegradable TiO2 nanowires for water treatment and remediation. They do not match in lifetime for a pratical use outside the lab testing experiments.

c) My point Q2. After reading your reply, I thank you for the information that the leaching is checked by ICP-MS when dealing with drinking water remediation. However, I maintain that I am not convinced by the answer : even for non drinkable water, such studies verifying that the new hybrid membranes are not leaching photocatlysts should be systematically performed. It would not be good to depollute waters from dyes BUT leach nanocatalysts into the environement. That was my point.

Reviewer 2 Report

The manuscript has been revised based on the comments and can be considered for the publication as it is.